

# Towards Cyber Physical Era: Soft Computing framework based Multi-Sensor Array for Water Quality Monitoring

Jyotirmoy Bhardwaj[1], Karunesh K. Gupta[1], Rajiv Gupta[2]

[1]Department of Electrical and Electronics Enginering, Birla Institute of Technology and Science, Pilani, 333031, India
[2]Department of Civil Enginering, Birla Institute of Technology and Science, Pilani, 333031, India
*Correspondence to*: Jyotirmoy Bhardwaj (Jyotirmoy.Bhardwaj@pilani.bits-pilani.ac.in)

**Abstract.**
New concepts and techniques are replacing traditional methods of water quality parameters measurement systems. This paper introduces the Cyber Physical System (CPS) approach for water quality assessment in distribution network. The proposed CPS consist of (a.) Sensing framework (b.) Soft Computing framework for computational modelling and data analysis. Sensing frame work consist of integrated Multi Sensor Array (MSA) and Soft Computing framework utilizes the applications of Python for user interface and Fuzzy Sciences for decision making. Extensive research suggest that pH, Dissolved Oxygen (D.O.), Electrical Conductivity (E.C.), Oxygen Reduction Potential (O.R.P.) and Temperature are suitable water quality parameters. Therefore, individual sensor of each parameter has been integrated to form Integrated MSA. Soft Computing mainly consist of Python framework for user interface and fuzzy logic for decision support system. Target of this proposed research is to provide simple, efficient, cost effective and socially acceptable means to detect the presence of contamination in water distribution network using applications of CPS.

## 1 Introduction

Water quality detection is paramount requirement before consumption by all human activities ranging from potable purpose to industrial and agriculture process. Water quality can be easily monitored in treatment plants and pumping stations but difficult to monitor in distribution network. However, once the treated water enters in distribution network, continuous real time monitoring becomes arduous task. Statistics show that 20-60% of water contamination incidents are related to events in the water distribution network. Application of sensors networking can sense and interact with water to identify the contamination in real time. Therefore, to detect water quality in distribution network, we propose implementation of Cyber Physical Systems (CPS) based water quality monitoring system to detect overall quality of water in distribution network. As with the advancements in Cyber-Physical Systems, the real time observation of environmental and physical systems can be effectively monitored and acted upon (Wang Zhui et al., 2015). The proposed CPS consist of a.) Sensing framework b.) Soft Computing framework for computational modelling and data analysis. In sensing framework, the MSA has been integrated from commercially available individual sensors of pH, Dissolved Oxygen (D.O.), Electrical Conductivity (E.C.), Oxygen Reduction Potential (O.R.P.) and Temperature.

In second stage, soft computing framework consist of microcontroller Arduino MEGA 2560 for MSA data processing, Python framework for user interface and Fuzzy logic for decision making. User Interface was developed using Python as, Python framework is effective tool that can handle low level and networking functionalities. Users are left only with task of application development as it reduces the dependency on expertise of embedded system in development of integrated sensor applications. Python has very clear syntax and offers an integrated development environment (Scherer D et. al, 2000). This framework enables interoperability & ease of integration and supports vision of Internet of Things (IoT). Finally, validation of system has been carried out through pre-processing of data points obtained from commercially available multiparameter detection system YSI Sonde 6820 V2 (YSI Y, 2013) and laboratory based analytical measurements.



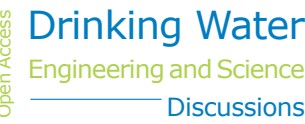

Table 1: Targeted Water Quality Parameters

| Parameters | Unit | Range for Potable water |
|---|---|---|
| pH | pH | 6.5-8.5 |
| Dissolved Oxygen | mg/lit | >3 mg/lit |
| Electrical Conductivity | uS/cm | 500-1000 |
| Temperature | ºC | 5-30 ºC |
| Oxygen Reduction Potential | mV | 650-800 |

## 2 System Design

### 2.1 Parameter Selection Strategy

US Environmental Protection Agency has carried out experimental evaluation of water sensors by analysing their performance on several different contaminations. The main conclusion was that chemical and biological contaminants generally affect pH, Dissolved Oxygen, Electrical Conductivity and Oxidation Reduction Potential (Lembrou et al., 2014). Therefore, the core idea was to detect the variations in above mentioned parameters that can significantly detect overall quality of water. Empirical evidences suggested that water quality parameters, such as pH, conductivity, and dissolved oxygen are sensitive indicators of nicotine, arsenic trioxide and Escherichia coli. Therefore, instead of direct detection of contaminants, monitoring these parameters is more feasible (Bhardwaj et al., 2015). Hence, we have selected five parameters and their range based on available scientific literature. Table 1 presents targeted Water Quality Parameters with widely acceptable ranges for potable water.

### 2.2 Sensing and soft computing Framework

The proposed sensing framework mainly consist of Multi Sensor Array (MSA) unit, Aurdino mega 2560 microcontroller unit, a Zigbee trans-receiver unit for wireless data transmission and reception. As discussed earlier MSA consist of five different types of sensor nodes mainly pH, Dissolved Oxygen (D.O.), Electrical Conductivity (E.C.), Temperature and Oxygen Reduction Potential (O.R.P.). Sensors draw power of 3-5 Volts for operation supplied by Arduino Mega 2560. The information sensed by the sensors then is converted into electrical signals, which is proportional to the actual values of parameters being sensed. The individual sensor units have been purchased from Atlas Scientific, USA as sensor nodes support open source architectures that further minimises the overall cost of the system. Table 2 presents summary of different sensor nodes used with their specifications and ranges. User interface has been developed through Python framework, widely accepted means for applications targeting sensor networks. This framework allow user to analyse the results in 2D and 3D space (David et. al, 2000) with potential benefits of open source community and wider programming choices. The applications of NumPy and Matplotlib libraries have been exploited while designing user interface (Ari and Mamatnazarova, 2006)

### 2.3 Decision Support System

Traditional methodologies cannot classify and quantify the targeted quantities, therefore soft computing approaches comes into scenario. Fuzzy sets or Fuzzy logic is one of the effective technique, can bolster findings of decision support system. Fuzzy approach allows translation of expression from natural language to mathematical universe and deals with ambiguity in decision analysis and already have been discussed in several environmental applications. Last but not least, it helps decision makers to make intelligent and well analysed decisions (Whittle, et al, 2006). For development of decision support system, three step approach have been formulated (a.) Identification of variables (b.) Determination of the Ranges of input and output variable (c.) Selection of the membership functions *(mf)* for various inputs and output.

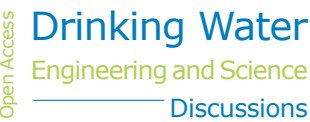

Table 2: Sensor Array Specification

| Parameters | Manufacturer | Range |
|---|---|---|
| pH | Atlas Scientific | 0-14 |
| Dissolved Oxygen | Atlas Scientific | 0-20 mg/lit |
| Electrical Conductivity | Atlas Scientific | 0.5 uS – 50,000 uS |
| Temperature | Atlas Scientific | -20 ºC to 130 ºC |
| Oxygen Reduction Potential | Atlas Scientific | +/- 2000 mV |

*mf* is generalisation of indicator function in classical sets and represents the extent of degree of validation. This technique has been widely accepted by researchers in applications, where decision making plays very important role (Zimmermann, 2012). The obtained membership function provides exact analysis of degree of validation of water quality in distribution network.

#### 2.3.1 DECISION SUPPORT SYSTEM: FUZZY APPROCH

Fuzzy set theory is a generalization of classical set theory, since the elements membership to the fuzzy set may be characterized in particular interval [0, 1]. Fuzzy set theory allows the integration of information from different parameters in the processes of modelling and evaluation. The *mf* represent the degree, or weighting, that the specified value belongs to the set (Zadeh, 1999). The membership function of the set A defined over a domain X takes the form of equation

$$\mu_A : X \rightarrow [0,1] \qquad\qquad (1)$$

The set *A* is defined in terms of its membership function by equation (2)

$$A = \{(\mu_A(X)), x \in X, \mu_A(X) \in [0,1]\} \qquad (2)$$

Fuzzy approach select Input variables set, has domain name called '*Universe of Discourse*' that are divided into subsets and expressed in linguistic terms. Fuzzy set operator will define '*if-then*' rules among the subsets of input, which will further define by *mf* of various forms such as trapezoidal, triangular, Gaussian etc. For this particular problem, inference rules has been applied which is mainly carried out by antecedent '*if*' part and consequent '*then*' part. The designed *mf* are triangular, although trapezoidal *mf* can also be used but triangular mf gives best response and quite easy to implement in fuzzy approach (Zhao and Bose, 2002). We have created membership functions based on fuzzy approach for pH, D.O., E.C., Temperature and O.R.P. The membership functions have been assigned for particular range of particular physiochemical parameter. Ranges of fuzzy sets used are shown in Table 3. The range selection of the physicochemical parameters to be monitored was based on extensive scientific literature on the relation between certain physicochemical parameters and chemical or biological contaminations that present in water (Lembrou et al., 2014).

Each physiochemical parameter has been assigned three membership functions that justify the suitable range of particular parameter for potable purpose. We selected three *mf* i.e Not Acceptable (NA), Adequate (ADE) & Highly Acceptable (HACC), & defined ranges for that particular function for each five water quality parameters. According to numerical ranges, the linguistic terms has been assigned to particular parameter. Figure 1 represent *mf* of pH and D.O. The rest of the mf can be referred through supplementary material. Defuzzification of mf has to be carry out, therefore, the

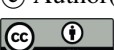



Table 3: Fuzzy Representation of Different Ranges

| Parameter | NA | ADE | HACC | NA |
|-----------|------|--------|----------|-------|
| pH | < 5.7 | 5.3-6.7 | 6.5-8.7 | >8.5 |
| D.O. | < 3 | 2.9-5.1 | 5.1-11.1 | >11 |
| E.C. | < 300 | 290-510 | 500-1050 | >1000 |
| O.R.P | < 550 | 530-670 | 650-820 | >800 |
| Temp. | < 2 | 1.9-10 | 9-36 | >35 |

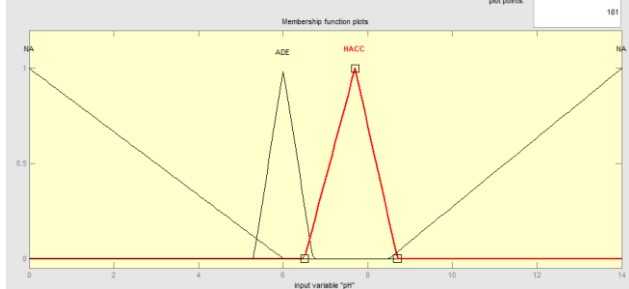
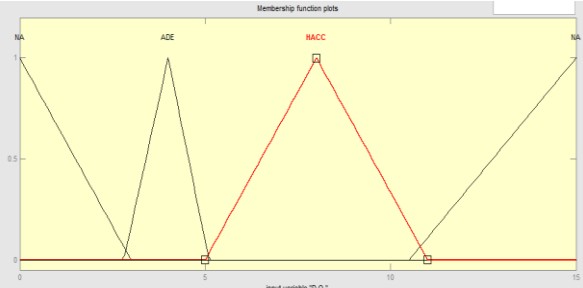

Figure 1. Membership function plot a.) pH and b.) Dissolved Oxygen

framework employ most prevalent Defuzzification approach, known as centroidal approach for Defuzzification. This approach is dominated by designing particular set of rules. In brief, refer to Rule No I "If the levels of pH in distribution network is Adequate (ADE), and the levels or Dissolved Oxygen (D.O.) are Highly Acceptable (HAC) then the resultant water quality is ADE". In fuzzy description, Rule I could be pronounce as follows "If pH is ADE, and D.O. is HAC then WQ is ADE". In case of NA of any of the parameter, the overall index result will be NA.

## 3. Experimental Procedure

The system has been deployed to five different water distribution sites of Birla Institute of Technology and Science, Pilani, India. The location has been selected at random to collect real time data in potable water distribution network. User Interface was created by novel python approach. The MSA was used to collect samples every second from potable water flowing through distribution network. The data transferred through ZigBee can be measured and analysed continuously with this user friendly approach. Primarily, the system has been designed to collect sensor data over an infinite period of time for real time continuous monitoring. Figure 2 represents real time collection of pH data and its user interface in python framework module. The results can be measured in real time or live with a gap of 1 second. Python graphical representation are presented over X-Y scale. X axis represents time in seconds (here we took 50 seconds period for analysis) and Y axis is respective parameters for each result. The pH value of this particular sample is 5.375, as confirmed by analytical analysis with error of +/- 0.02 value. The dots represent time at which values are taken and for convenience and limitations, the first 50 second values are presented.



Table 4 The Average values of collected samples.

| Location | pH | | | D.O. (mg/lit) | | | E.C. (uS/cm) | | | O.R.P. (mV) | | | Temp. (ºC) | | |
|---|---|---|---|---|---|---|---|---|---|---|---|---|---|---|---|
| | MSA | YSI | Lab. | MSA | YSI | Lab. | MSA | YSI | Lab. | MSA | YSI | Lab. | MSA | YSI | Lab. |
| 1 | 7.17 | 7.12 | 7.10 | 7.80 | 8.20 | 8.00 | 360 | 385 | 390 | 680 | 677 | 660 | 23.7 | 23.1 | 23 |
| 2 | 7.62 | 7.66 | 7.60 | 6.30 | 7.20 | 7.45 | 440 | 460 | 500 | 700 | 720 | 690 | 19.6 | 20.0 | 20 |
| 3 | 7.43 | 7.50 | 7.40 | 9.50 | 9.20 | 9.30 | 507 | 490 | 505 | 720 | 729 | 709 | 24.6 | 24.4 | 25 |
| 4 | 7.26 | 7.22 | 7.20 | 8.80 | 9.00 | 9.70 | 390 | 380 | 400 | 750 | 742 | 737 | 19.1 | 19.2 | 19 |
| 5 | 6.90 | 6.93 | 6.90 | 6.90 | 6.93 | 6.90 | 340 | 365 | 350 | 660 | 670 | 680 | 18.2 | 18.1 | 18 |

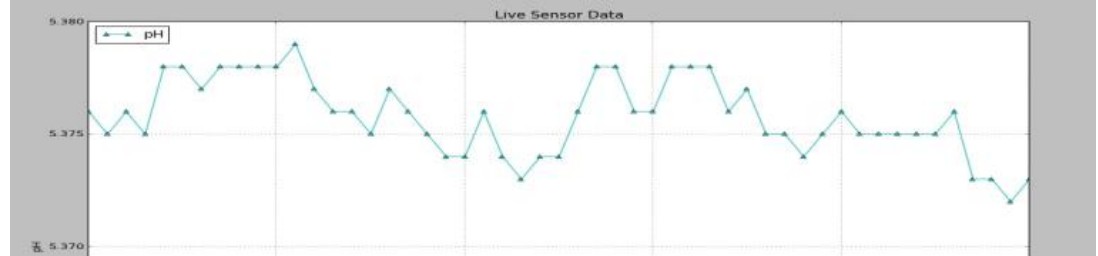

Figure 2. Real Time Data Collection from Python Framework Module for pH

### 3.1 Validation & Performance Comparison

As mentioned earlier, the system, consist of Integrated MSA of five individual sensors, Arduino MEGA 2560 with Soft Computing framework has been deployed to five different locations of Birla Institute of Technology and Science, Pilani Campus. The validation of the proposed system was carried out using commercially available multiparameter system YSI SONDE 6820. Water samples have been collected from five different locations in potable Water distribution network. The targeted parameters were measured analytically and thereafter compared by proposed system and commercially available YSI Sonde 6820 multiparameter water quality measurement system. All the five parameters were measured analytically with the traditional lab based measurement technique. pH was measured by Titration with sodium hydroxide method, D.O was measured with Winkler method, E.C. was determined by two flat cylindrical electrodes, O.R.P was measured by inert sensing electrode in contact with the solution and a stable reference electrode connected to the solution by a salt bridge, and temperature was measured by lab based water temperature sensor. Baud rate of system was 9600; Data rate 1Kbps without parity and Kermit protocol has been applied for YSI Sonde 6820 V2 System. Calibration of overall system was carried out through standard available buffer solutions of different pH of 4.0, 7.0 & 10.0, E.C. node was calibrated by 80,000 uS/cm and 12880 uS/cm solution, D.O. node was calibrated by known D.O. solution of 8 mg/lit and O.R.P node was calibrated with 225 mV solution. Proposed system was compared by collecting samples from five different locations of distribution network.

Samples of these models are compared with experimental results and are illustrated in figure 3 and the average value to the obtained samples are highlighted in Table 4. In Figure 4, Number 1-5 represent different location used for comparison. The Blue, Red and Green line represents MSA, YSI Sonde 6820 V2 and analytical results respectively. The Comparative analysis of different parameters clearly points out degree of variations of different parameters measured at different location. The performance of proposed system is compatible and with analytical and YSI Sonde 6820 V2 System.



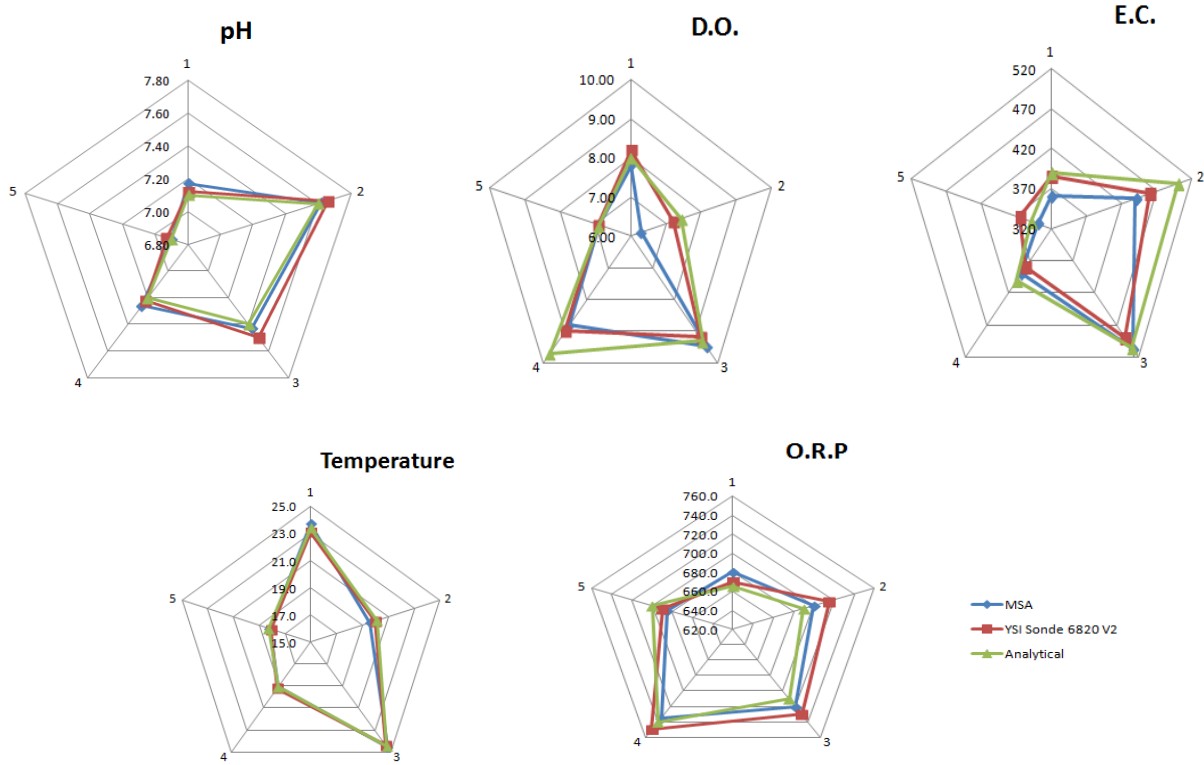

Fig 3: Comparative analysis of Each Parameter of MSA with YSI Sonde & Analytical Laboratory Results on locations 1-

**Conclusion**

The latest advancements in information and communication technologies are expected to capture relevant data without any ambiguity and delay. In so far, traditional and analytical techniques have been used, which are, however, irrelevant for contamination event detection in distribution network. Cyber Physical Systems are making sensing of environmental applications ubiquitous, with advanced capability of huge data processing and thus can improve decision making with respect to many poorly understood water issues. Since reliable and effective continuous water quality monitoring has always been challenging, therefore this paper demonstrated the CPS system based water quality monitoring system in distribution network.

This method is an effective way to detect water quality in distribution network compared to manual approaches. Moreover, data acquired by this prototype is user friendly will assist in better understanding of water distribution network and can be useful to interpret more valuable information by analyzing interrelationship among physiochemical parameters using techniques like ANN, PLS etc. This paper demonstrates the integration of soft computing techniques Python and Fuzzy logic with Integrated MSA to facilitate decision support system. Comparative analysis of developed system with analytical measurement techniques and commercially available system has been carried out. On analyzing both models at different locations, it can be inferred that their performance is being governed by the robustness of



integrated sensor. The proposed system can be implemented in remote locations and unlike commercially available analyzers, the developed system is low cost, low power, lightweight and capable to process, log, and remotely present data.

**Acknowledgements.**
The development of this system would not have been possible without the support of Centre of Excellence, in Waster, Water and Energy Management, Birla Institute of Technology and Science, Pilani. We would particularly like to thank Professor Rajiv Gupta for valuable inputs and far-sighted approach. We are also thankful to Professor Ramón Martínez Mañez, Instituto Interuniversitario de Investigación de Reconocimiento Molecular y Desarrollo Tecnológico,
Universidad Polytechnica de Valencia, Spain for valuable training and support.

**Competing interests**. The authors declare that they have no conflict of interest.

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
