# Peer review of "Towards Cyber Physical Era: Soft Computing framework based Multi-Sensor Array for Water Quality Monitoring"

_Drinking Water Engineering and Science, 2017_

## Referee Comment (RC1) · Anonymous Referee #1 · 25 Aug 2017

The manuscript introduces a water quality monitoring system with which data of pH, DO, EC, ORP, and temperature are measured by a sensor array and analyzed by a Python program using fuzzy set theory. The authors are invited to consider the following comments:

GENERAL COMMENTS:

1. Section 2.3.1 proposed a fuzzy approach for the water quality decision support system. However, in Section 3 there is little result analysis or discussion about the results of applying fuzzy logic to the collected live data. The code included in the supplemental materials does not show the application of fuzzy logic, either.

Readers would benefit from descriptive text and/or figures showing how the fuzzy approach classifies water quality data at the 5 sites into the 3 proposed categories of NA, ADE, and HACC, and what advantages the fuzzy approach has over other algorithmic classifiers or empirical methods.

2. The Conclusion section reads "The proposed system can be implemented in remote locations and unlike commercially available analyzers, the developed system is low cost, low power, lightweight and capable to process, log, and remotely present data." However, the article does not compare the (estimated) costs, power consumption, or weights, of the proposed system and commercial systems.

Other Comments:

1. Page 1, Line 17. "socially acceptable means to detect... contamination". Suggest clarifying the meaning of "social acceptance" in the context of water quality monitoring systems.

2. Page 1, Line 24-25. "Statistics show that 20-60% of water contamination incidents are related to events in the water distribution network". References are needed here.

3. Table 1 shows that for DO values, range for portable water is " >3mg/L". However, if groundwater is used as water source, regular DO values are usually lower than 3 mg/L. See, e.g., Sarin, P., et al (2004). Same comments for the DO ranges in Table 3.

4. Page 2, Line 19. "Aurdino mega 2560 microcontroller...". is this "Arduino"?

5. Table 2. suggest removing the column of "manufacturer". It has been mentioned in the main text.

6. Page 3, Line 45. Suggest numbering the supplemental materials and refer to them in the text by numbers.

7. Page 4. Line 10-14. Are there only two rules? if not, suggest listing all rules.

8. Page 4. Table 4. suggest adding a column showing the Mean Average Percentage

Error (MAPE) of measurements for the proposed system versus lab results.

9. Page 6, Line 18. What does PLS stands for? Abbreviations should be spelled out on first occurrence.

10. How long did the system run at the 5 sites? how many measurements are taken in total? How was the system powered on site? It may also be beneficial to include raw data in the supplemental materials.

Comments on the figures:

Figure 1. The text size is too small to read. Suggest increasing the size and removing the grey background.

Figure 2. X-axis is not shown.

Figure 3. Radar chart usually shows different types (rather than sites) of measurements on the axes (See Figure 5, Lambrou et. al. 2014). Suggest re-plotting to have 5 charts for 5 sites.

Reference:

Sarin, P., et al. "Iron release from corroded iron pipes in drinking water distribution systems: effect of dissolved oxygen." Water Research 38.5 (2004): 1259-1269.

---

## Referee Comment (RC2) · Anonymous Referee #2 · 21 Sep 2017

The authors present an online water quality monitoring and contamination detection system for water distribution networks. Data processing and graphical user interface is implemented by use of Python libraries. For detection of contamination events Fuzzy set theory is applied. The topic is interesting. However, before publication the manuscript needs major revision.

The following issues should be addressed by the authors:

- The quality of language of the document has to be generally improved. Sometimes the meaning is not clear. Examples: Line 35 f. "User Interface was developed using Python as, Python framework is effective tool that can handle low level and networking

functionalities."

Many statements are not specific enough Example: Line 33f: "Traditional methodologies cannot classify and quantify the targeted quantities, therefore soft computing approaches comes into scenario." Line 39f: "This framework enables interoperability & ease of integration and supports vision of Internet of Things (IoT)." Python is a script language. Refer to specific library

- The authors claim that the "Target of this proposed research is to provide simple, efficient, cost effective and socially acceptable means to detect the presence of contamination in water distribution network using applications of CPS." However, social acceptance is not addressed in the paper.

- Similarly, it is not explained why the approach is cost effective. Cost for data transfer by wireless technology, maintenance of the system. For real time application: Frequency of data transfer has impact on battery life time, cost, storage capabilities, data treatment.

- Use of Fuzzy set technology is not well explained. Comparison with existing techniques (PCA: Principal component analysis, . . .) is missing. Why is Fuzzy theory superior?

- "User friendly Interface" -> more details required.

- Information about detection capabilities is missing: reliability of detection, rate of false positive detections

---

## Short Comment (SC1) · 5 Oct 2017

1. If (pH is NA) **AND** (D.O. is NA) **AND** (E.C. is NA) **AND** (O.R.P is NA) **AND** (Temperature is NA) then (Water Quality is NA)
2. If (pH is ADE) **AND** (D.O. is AND) **AND** (E.C. is ADE) **AND** (O.R.P is ADE) **AND** (Temperature is ADE) then (Water Quality is ADE)
3. If (pH is HACC) **AND** (D.O. is HACC) **AND** (E.C. is HACC) **AND** (O.R.P is HACC) **AND** (Temperature is HACC) then (Water Quality is HACC)
4. If (pH is ADE) **OR** (D.O. is not NA) **OR** (E.C. is not NA) **OR** (O.R.P is not NA) **OR** (Temperature is not NA) then (Water Quality is ADE)
5. If (pH is not NA) **OR** (D.O. is ADE) **OR** (E.C. is not NA) **OR** (O.R.P is not NA) **OR** (Temperature is not NA) then (Water Quality is ADE)
6. If (pH is not NA) **OR** (D.O. is not NA) **OR** (E.C. is ADE) **OR** (O.R.P is not NA) **OR** (Temperature is not NA) then (Water Quality is ADE)
7. If (pH is not NA) **OR** (D.O. is not NA) **OR** (E.C. is not NA) **OR** (O.R.P is ADE) **OR** (Temperature is not NA) then (Water Quality is ADE)
8. If (pH is not NA) **OR** (D.O. is not NA) **OR** (E.C. is not NA) **OR** (O.R.P is not NA) **OR** (Temperature is ADE) then (Water Quality is ADE)
9. If (pH is HACC) **OR** (D.O. is not NA) **OR** (E.C. is not NA) **OR** (O.R.P is not NA) **OR** (Temperature is not NA) then (Water Quality is ADE)
10. If (pH is not NA) **OR** (D.O. is HACC) **OR** (E.C. is not NA) **OR** (O.R.P is not NA) **OR** (Temperature is not NA) then (Water Quality is ADE)
11. If (pH is not NA) **OR** (D.O. is not NA) **OR** (E.C. is HACC) **OR** (O.R.P is not NA) **OR** (Temperature is not NA) then (Water Quality is ADE)
12. If (pH is not NA) **OR** (D.O. is not NA) **OR** (E.C. is not NA) **OR** (O.R.P is HACC) **OR** (Temperature is not NA) then (Water Quality is ADE)
13. If (pH is not NA) **OR** (D.O. is not NA) **OR** (E.C. is not NA) **OR** (O.R.P is not NA) **OR** (Temperature is HACC) then (Water Quality is ADE)

Note: The summary of above rules can be refer from Section 2.3.1 of research paper.

---

## Short Comment (SC2) · 5 Oct 2017

I have read the manuscript. However, the findings do not reflect the cost analysis and comparisons of overall system. Kindly mention

a. The overall system cost and comparison analysis ? b. Does the proposed system is feasible for rural deployments ? c. Is is possible to measure fluoride contamination, as author claim that system is adaptable ? d. Is this proposed system can be integrated with any other sensor array for water?

25, 2017.

---

## Author Comment (AC1) · 5 Oct 2017

For benefit of readers, the authors would like to incorporate additional figures and supplementary file.

Fig 1. Represents System Layout and Prototype, Fig 2. Representation of Fuzzy Rule, Fig 3 Represents Steps of Measurement Cycle and Fig 4 Represents Mean Average Percentage Error chart of SA with respect to YSI V2 and analytical results. The supplementary file presents application of Fuzzy inference through Python.

Please also note the supplement to this comment:

[Figure]

https://www.drink-water-eng-sci-discuss.net/dwes-2017-25/dwes-2017-25-AC1-supplement.pdf

[Figure]

(a)

[Figure]

(b)

[Figure]

5  Fig1. (a) Design stages for proposed system (b) System Prototype

[Figure]

5   Figure 3. Representation of Fuzzy Rule

Figure 3 is the representation of the rule
"If (pH is ADE) OR (D.O. is HACC) OR (E.C. is ADE) OR (O.R.P is HACC) OR (Temperature is HACC) then (Water
10   Quality is ADE)".

[Figure]

Fig 4. Steps of Measurement Cycle

[Figure]

Fig 6: MAPE Chart (in %) (a) SA against Laboratory Results (b) SA against YSI Sonde 6820 V2 results

5    Table 4 MAPE (in %) Table

|   | pH | | D.O. (mg/lit) | | E.C. (uS/cm) | | O.R.P. (mV) | | Temp. (ºC) | |
|---|-----|-----|-----|-----|-----|-----|-----|-----|-----|-----|
|   | YSI | Lab | YSI | Lab | YSI | Lab | YSI | Lab | YSI | Lab |
| 1 | 0.702 | 0.986 | 4.878 | 2.500 | 6.494 | 7.692 | 0.443 | 3.030 | 2.597 | 3.043 |
| 2 | 0.522 | 0.263 | 1.389 | 2.013 | 4.348 | 9.651 | 2.778 | 1.449 | 2.000 | 2.000 |
| 3 | 0.933 | 0.405 | 3.261 | 2.151 | 3.469 | 0.396 | 1.235 | 1.551 | 0.820 | 1.600 |
| 4 | 0.554 | 0.833 | 2.222 | 3.297 | 2.632 | 2.500 | 1.078 | 1.764 | 0.521 | 0.526 |
| 5 | 0.433 | 0.000 | 0.433 | 0.000 | 4.762 | 2.857 | 1.493 | 2.941 | 0.552 | 1.111 |

**Supplement:**

```python
**============================================**
**Fuzzy Systems: Water Quality Monitoring**
**============================================**

"""
import numpy as np
import skfuzzy as fuzz
from skfuzzy import control as ctrl
**THe process of obtaining imput has been explained in Supplementary_Material_2.pdf**
**Antecedent/Consequent for water quality monitoring**
**functions**
pH = ctrl.Antecedent(np.arange(0, 14, 1), 'pH')
DO = ctrl.Antecedent(np.arange(0, 12, 1), 'DO')
EC = ctrl.Antecedent(np.arange(0, 1100, 50), 'EC')
ORP = ctrl.Antecedent(np.arange(0, 1100, 50), 'ORP')
Temperature= ctrl.Antecedent(np.arange(0, 40, 2.5), 'Temperature')
Water_Quality = ctrl.Consequent(np.arange(0, 10, 1), 'Water_Quality')
**Generate Fuzzy membership function for antecedent**
pH['NA'] = fuzz.trimf(pH.universe, [0, 0, 5.7])
pH['ADE'] = fuzz.trimf(pH.universe, [2.9, 4.0, 5.1])
pH['HACC'] = fuzz.trimf(pH.universe, [6.5, 7.6, 8.7])
pH['NA'] = fuzz.trimf(pH.universe, [8.5, 14, 14.1])
DO['NA'] = fuzz.trimf(DO.universe, [0, 0, 3])
DO['ADE'] = fuzz.trimf(DO.universe, [5.3, 6, 6.7])
DO['HACC'] = fuzz.trimf(DO.universe, [5.1, 8, 11.1])
DO['NA'] = fuzz.trimf(DO.universe, [11, 12, 12.1])
EC['NA'] = fuzz.trimf(EC.universe, [0, 0, 300])
EC['ADE'] = fuzz.trimf(EC.universe, [290, 400, 510])
EC['HACC'] = fuzz.trimf(EC.universe, [650, 740, 820])
EC['NA'] = fuzz.trimf(EC.universe, [800, 1000, 1100])
ORP['NA'] = fuzz.trimf(ORP.universe, [0, 0, 550])
ORP['ADE'] = fuzz.trimf(ORP.universe, [530, 600, 670])
ORP['HACC'] = fuzz.trimf(ORP.universe, [650, 740, 820])
ORP['NA'] = fuzz.trimf(ORP.universe, [800, 1000, 1100])
Temperature['NA'] = fuzz.trimf(Temperature.universe, [0, 0, 2])
Temperature['ADE'] = fuzz.trimf(Temperature.universe, [1.9, 5, 10])
Temperature['HACC'] = fuzz.trimf(Temperature.universe, [9, 21, 36])
Temperature['NA'] = fuzz.trimf(Temperature.universe, [35, 37.5, 40])
**Custom membership function for antecedent**
**Pythonic API**
Water_Quality['NA'] = fuzz.trimf(Water_Quality.universe, [0, 0, 4])

Water_Quality['ADE'] = fuzz.trimf(tip.universe, [4, 5.5, 7])
Water_Quality['HACC'] = fuzz.trimf(tip.universe, [7, 8.5, 10])
**You can see how these look with .view()for example**
pH['ADE'].view()
**Fuzzy Rule Generation (For Convineance only one rule have been shown here, rest of the rules can be referred from supplementary material and can be implemented**
**through same fashion**
**If (pH is ADE) OR (D.O. is HACC) OR (E.C. is HACC) OR (O.R.P is HACC) OR (Temperature is ADE) then (Water Quality is ADE)**
rule1 = ctrl.Rule(pH['ADE'] | DO['HACC'] | EC['HACC'] | ORP['HACC'] | Temperature['ADE'], Water_Quality['ADE'])
rule1.view()
**To create a control for one rule, However to change the rules control has to change**
WaterQ= ctrl.ControlSystem([rule1])
**Simulate control**
WaterQ = ctrl.ControlSystemSimulation(Water_Quality_ctrl)
**Input Process can be understood from Supplementary_Material_2, However, Input have to be averaged before supply to Fuzzy framework.**
**To understand the process let consider arbitrary values**
WaterQ.input['pH'] = 5.95
WaterQ.input['DO'] = 8.08
WaterQ.input['EC'] = 406
WaterQ.input['ORP'] = 735
WaterQ.input['Temperature'] = 22.8
WaterQ.compute()
print WaterQ.output['Water_Quality']
Water_Quality.view(sim=WaterQ)
```

---

## Author Comment (AC2) · 6 Oct 2017

We are thankful to Reviewer_1 for constructive comments, which helped us to improve the overall quality of manuscript. To address the reviewer concerns, we have made significant changes in section 2 and section 3 of manuscript. For convenience, the reviewer's comments are kept as it is, while author response are kept in italics.

GENERAL COMMENTS:
1. Section 2.3.1 proposed a fuzzy approach for the water quality decision support system. However, in Section 3 there is little result analysis or discussion about the results of applying fuzzy logic to the collected live data. The code included in the supplemental materials does not show the application of fuzzy logic, either. Readers would benefit from descriptive text and/or figures showing how the fuzzy approach classifies water quality data at the 5 sites into the 3 proposed categories of NA, ADE, and HACC, and what advantages the fuzzy approach has over other algorithmic classifiers or empirical methods.

   a. [In Section 3 there is little result analysis or discussion about the results of applying fuzzy logic to the collected live data. The code included in the supplemental materials does not show the application of fuzzy logic, either.

   *Reply*
   *In compliance with reviewers comment, the following amendments have been carried out.*

      o *In Section 2.3, Text of Decision Support System has been improved with additional details and the fuzzy Rules have been clarified in revised manuscript. Similarly, the approach of applying fuzzy inference have been depicted in additional figure no 3 of revised manuscript. Please refer figure number 3 in author Comment (AC 1) section that demonstrate how to apply fuzzy rules over data points. Additional supplement material have been added in short Comment (SC1) with file that mention all the fuzzy rules. As far as code is concerned, the Python code utilizing the functionalities of Skfuzzy library has been included in supplementary material (Link), which shows the application of fuzzy through Python.*

   b. [Readers would benefit from descriptive text and/or figures showing how the fuzzy approach classifies water quality data at the 5 sites into the 3 proposed categories of NA, ADE, and HACC]
   *Additional descriptive text has been introduced in revised manuscript. Moreover, additional figure has been added to show the interpretation of fuzzy rule (refer AC1 figure 3, or refer fig 3 of revised manuscript). This figure depicts, how fuzzy approach classify water quality and support decision making.*

   c. [and what advantages the fuzzy approach has over other algorithmic classifiers or empirical methods.]
   o *Additional text has been added in revised manuscript with Section 2.3. This particular text paragraph demonstrate the superiority of fuzzy over PCA and other regression methods. The text is mentioned below is part of revised manuscript with additional references.*

   "Literature review indicates, fuzzy perform better than both linear and non-linear regression methods in terms of model building, adaptive modelling and decision making (Doorsy and Coovert, 2003). Although, Principal Component Analysis (PCA) is also one of the favorite tool for information extraction and analysis. However, PCA is sensitive to missing data and poor correlation among water quality parameters (Sarbu and Pop, 2005). Moreover, fuzzy offers simplicity, flexibility, reliable results, can handle incomplete data sets and nonlinear functions. Therefore, Fuzzy has been extensively used in development of decision support system for applications pertaining to water and CPS. This approach have been widely discussed in several environmental applications ranging from development of decision support system based for 45 urban water management (Macropoulas et al., 2003) to Fuzzy based CPS system (Leu and Zhang, 2009)".

   o *Similarly, Additional text in section 2.3 included demonstrating comparison of Python with MATLAB in context of manuscript.*

2. The Conclusion section reads "The proposed system can be implemented in remote locations and unlike commercially available analyzers, the developed system is low cost, low power, lightweight and capable to process, log, and remotely present data." However, the article does not compare the (estimated) costs, power consumption, or weights, of the proposed system and commercial systems.

   *Reply:*

   (a) [However, the article does not compare the (estimated) costs of the proposed system and commercial systems]
   *Additional text, Section 2.4 Comparative Analysis of System Cost has been added to revised manuscript to present estimated cost and comparison. The text is as follows:*

   "Commercially available multiparameter water quality monitoring system (eg. YSI Sonde V2) varies in the range of 5000 US $ to 8000 US $ (with computing framework) mainly used for Industrial purpose. On the other side, general purpose sensor nodes of commercially available Vernier cost around 800 US$ to 1000 US$ (without computing framework) for potable water testing. The cost of commercially available computing tools (eg. MATLAB and LoggerPro) varies in the range of 350 US$ to 500 US$. By exploiting the benefits of open source computing modules and libraries, the overall system cost can significantly be lower down. For proposed system, the cost of sensor array is summation of individual cost of pH, DO, ORP, EC and Temperature nodes and was 530 US$. In addition, the hardware platform has a cost of 59 US$, which includes Arduino MEGA 2560 and XBee (wireless data transmission unit). Therefore, overall cost of sensors and hardware unit was 589 US $. The cost of consumables, data collection, power source, scientific supervision, labor, resources used for sample collection and shipping to analytical laboratories has not been taken into account, as it will be approximately same for all other commercially available systems".

   (b) [For power consumption and weight]
   *The power consumption and weight has been mentioned in revised manuscript section 2.2. We have decided to opt for light weight hardware platform unit with low power requirements of 5-15 V. The Sensor Array operates on the power ranging from 5-12 V, therefore low power platform unit is sufficient for overall operations and simultaneously can be supported by Li-Ion batteries.*

Other Comments:

1. Page 1, Line 17. "socially acceptable means to detect... contamination". Suggest clarifying the meaning of "social acceptance" in the context of water quality monitoring systems.
   *Reply: Socially acceptable term has been removed from the revised manuscript, as Social Acceptance is wide ranging term outlining societal cooperation and contribution to various other economic factors. Since, the proposed paper do not cover such issues, except cost analysis. Therefore, this term "socially acceptable" has been removed from the text.*

2. Page 1, Line 24-25. "Statistics show that 20-60% of water contamination incidents are related to events in the water distribution network". References are needed here.
   *Reply: Authors are thankful to reviewer to rectify this mistake. The line has been removed from the text of revised manuscript.*

3. Table 1 shows that for DO values, range for portable water is " >3mg/L". However, if groundwater is used as water source, regular DO values are usually lower than 3 mg/L. See, e.g., Sarin, P., et al (2004). Same comments for the DO ranges in Table 3.
   *Reply: Very low D.O. is indicative of too many bacteria and excess BOD. On the other hand very high D.O. corroborated the rate of corrosion in distribution pipes leading to contamination. In Contrast, Groundwater at excess depth may have D.O. < 3mg/lit even without being contaminated. Since D.O is variable with temperature, therefore, from groundwater extraction to distribution the levels of D.O may vary. Still, it should not be very low for human consumption, as in general sense very low D.O. is indicative of water contamination.*
   *Reference: Causes of low Dissolved Oxygen (https://www.pca.state.mn.us/sites/default/files/wq-iw3-24.pdf)*

4. Page 2, Line 19. "Aurdino mega 2560 microcontroller...". Is this "Arduino"?
*Reply: Arduino (Mistake Rectified)*

5. Table 2. Suggest removing the column of "manufacturer". It has been mentioned in the main text.
*Reply: In compliance with suggestions, we have removed the column.*

6. Page 3, Line 45. Suggest numbering the supplemental materials and refer to them in the text by numbers.
*Reply: All Supplementary material file names has been renamed and now referred by numbers in text of revised manuscript.*

7. Page 4. Line 10-14. Are there only two rules? if not, suggest listing all rules.
*Reply: Rules have been clarified in revised manuscript. Additional supplement material have been added with file name 4. Supplumentary_Material_4.pdf. Moreover, additional text has been added to improve the clarity of rules in section 2.3.*

8. Page 4. Table 4. Suggest adding a column showing the Mean Average Percentage Error (MAPE) of measurements for the proposed system versus lab results.
*Reply: Extra table have been added for MAPE for all the five parameters with graphical representation. Utmost care has been taken while collecting the samples from SA (Sensor Array) and Laboratory samples. SA incorporate Industry manufactures sensors, and we do not have any control over their values except repeated calibrations from already known samples. Please refer fig 6 in Author Comment 1 that depict MAPE of SA. We have also included the same MAPE figure in revised manuscript in place of Radar Chart.*

9. Page 6, Line 18. What does PLS stands for? Abbreviations should be spelled out on first occurrence.
*Reply: PLS stands for Partial Least Square. The mistake has been rectified in revised manuscript with all the abbreviations.*

10. How long did the system run at the 5 sites? how many measurements are taken in total? How was the system powered on site? It may also be beneficial to include raw data in the supplemental materials.
*Reply: Additional paragraph has been added in revised manuscript labelled as section 3.1.2 to mention the procedure, numbers of cycles and time duration for the system. The power method is mentioned in section 2.1.2 of revised manuscript. As far as raw data is concerned, we have generated thousands of Data points through different iterations, which is not possible to include due to huge numbers. However, we will include file named Supplementary_material_7.pdf for to show the results of one iterations.*

Comments on the figures:
Figure 1. The text size is too small to read. Suggest increasing the size and removing the grey background.
*Reply: Grey background has been removed and text size has been improved.*

Figure 2. X-axis is not shown.
*Reply: Shown*

Figure 3. Radar chart usually shows different types (rather than sites) of measurements on the axes (See Figure 5, Lambrou et. al. 2014). Suggest re-plotting to have 5 charts for 5 sites.
*Reply: Earlier Radar charts have been removed and replaced by MAPE chart for all the five locations. Please Refer figure 6 in AC 1 for MAPE calculations.*

---

## Author Comment (AC3) · 6 Oct 2017

We are thankful to Reviewer_2 for constructive comments, which helped us to improve the overall quality of revised manuscript. To address the reviewers concerns, we have made significant changes in section 2 and section 3 of manuscript. Please find the author response in italics for reviewer's comments.

Comments of Reviewer_2

1. The quality of language of the document has to be generally improved. Sometimes the meaning is not clear. Examples: Line 35 f. "User Interface was developed using Python as, Python framework is effective tool that can handle low level and networking functionalities."

    (a) [The quality of language of the document has to be generally improved]
    *Reply: The language has been considerably improved in revised manuscript. To clear the meaning, additional text has been introduced in Revised Manuscript.*
    o *Section 2.2, Improved Explanation of Platform design.*
    o *Section 2.3, Improved Explanation of Decision Support System*
    o *Additional Figures has been introduced (Refer fig in Author Comment 1) and quality of rest of the figures has been significantly improved.*
    o *Similarly, the fuzzy description has been improved in manuscript with rules. Please refer section 2.3.1 of revised manuscript.*
    o *Literature review for scientific reasoning has been introduced in revised manuscript with additional references. Please refer following text as part of revised manuscript with additional references.*

    "The python module is an effective tool to reduce the complexity of overall system, usually deployed at client side as this allows user to analyse the results in 2D/3D space in user friendly way (Scherer et al., 2000). In addition offers benefits of open source community and wider programming choices. Python module has been instrumental in development of software architecture framework to behavioral modelling for CPS (Ringert et al., 2014). On the other hand, in the development of CPS test-beds, the python module has been used as it supports adaptability and re-configurability (Adhikari et. al, 2016). Although, MATLAB is also potential choice for development of soft computing framework for CPS. However, Python offers advantages over MATLAB mainly due to Open Source with comprehensive library, choices of 2D/3D graphic packages, ease of re-configurability and low cost."

    (b) [Sometimes the meaning is not clear. Examples: Line 35 f. "User Interface was developed using Python as, Python framework is effective tool that can handle low level and networking functionalities]
    *Reply: Sentences ambiguity has been sorted out in revised manuscript. The statement has been replace with* "The python module is an effective tool to reduce the complexity of overall system, usually deployed at client side as this allows user to analyse the results in 2D/3D space in user friendly way (Scherer et al., 2000). In addition offers benefits of open source community and wider programming choices."

2. Many statements are not specific enough Example: Line 33f: "Traditional methodologies cannot classify and quantify the targeted quantities, therefore soft computing approaches comes into scenario." Line 39f: "This framework enables interoperability & ease of integration and supports vision of Internet of Things (IoT)." Python is a script language. Refer to specific library

    Reply:
    (a) *The overall language has been significantly improved in revised manuscript. The sentence begin with "Traditional…" has been removed from the text. Whereas, the sentence begin with "This Framework…" has been modified and new term reconfigurable has been introduced in the text. Please refer text* "The re-configurability and scalability offers value addition, as it offer freedom to modify the system as per changing application requirements and improves the adaptability of overall system in different

scenarios. CPS are primarily scalable and reconfigurable systems and can be modified based on volume of data, bandwidth requirements, power requirements and sensing applications."

[Python is a script language. Refer to specific library]

(b) *The specific libraries for Python has already been mentioned in discussion manuscript. Please refer page no 2, line no 29. Moreover, these libraries has been defined in supplementary material (please refer python code in supplementary material)*

3. The authors claim that the "Target of this proposed research is to provide simple, efficient, cost effective and socially acceptable means to detect the presence of contamination in water distribution network using applications of CPS." However, social acceptance is not addressed in the paper.

*Reply:*

(a) ["Target of this proposed research is to provide simple and cost effective…."]
*Additional text in section 2.4 depicts the comparative analysis for cost effective ness. The text is as follows*
*"Commercially available multiparameter water quality monitoring system (eg. YSI Sonde V2) varies in the range of 5000 US $ to 8000 US $ (with computing framework) mainly used for Industrial purpose. On the other side, general purpose sensor nodes of commercially available Vernier cost around 800 US$ to 1000 US$ (without computing framework) for potable water testing. The cost of commercially available computing tools (eg. MATLAB and LoggerPro) varies in the range of 350 US$ to 500 US$. By exploiting the benefits of open source computing modules and libraries, the overall system cost can significantly be lower down. For proposed system, the cost of sensor array is summation of individual cost of pH, DO, ORP, EC and Temperature nodes and was 530 US$. In addition, the hardware platform has a cost of 59 US$, which includes Arduino MEGA 2560 and XBee (wireless data transmission unit). Therefore, overall cost of sensors and hardware unit was 589 US $. The cost of consumables, data collection, power source, scientific supervision, labor, resources used for sample collection and shipping to analytical laboratories has not been taken into account, as it will be approximately same for all other commercially available systems".*

(b) [However, social acceptance is not addressed in the paper.]
*Socially acceptable term has been removed from the text, as Social Acceptance is wide ranging term outlining societal cooperation, contribution to various other economic factors. Since, the proposed paper do not cover such issues, except cost analysis. Therefore, this term "socially acceptable" has been removed from the text.*

4. Similarly, it is not explained why the approach is cost effective. Cost for data transfer by wireless technology, maintenance of the system. For real time application: Frequency of data transfer has impact on battery life time, cost, storage capabilities, and data treatment.

*Reply:*

(a) [For why the approach is cost effective?]
*Please refer Reply 3. The explanation of cost effective has been included.*

(b) [For Cost for data transfer by wireless technology, maintenance of the system?]
*Cost of wireless data transmission has been mentioned on hardware platform design. However, maintenance cost has not been included. Please refer text in reply 3.*

5. Use of Fuzzy set technology is not well explained. Comparison with existing techniques (PCA: Principal component analysis, . . .) is missing. Why is Fuzzy theory superior?

*Reply:*

(a) [Use of Fuzzy set technology is not well explained?]

*The text relevant to fuzzy logic have been significantly improved in revised manuscript in terms of overall language, rules development and procedure. The principle on which rules are based have been improved. Additional figure to describe the rule procedure has been added to text (refer fig 3 in Author Comment Section 1 and Short Comment Section 2). Literature review has been included and Supplementary file has been included by name Supplementary_Material_4.pdf with all the rules mentioned.*

*(b)* [Comparison with existing techniques (PCA: Principal component analysis) is missing. Why is Fuzzy theory superior?]
*Additional text has been added in revised manuscript with Section 2.3. This particular text paragraph demonstrate the superiority of fuzzy over PCA and other regression methods.*

"Literature review indicates, fuzzy perform better than both linear and non-linear regression methods in terms of model building, adaptive modelling and decision making (Doorsy and Coovert, 2003). Although, Principal Component Analysis (PCA) is also one of the favorite tool for information extraction and analysis. However, PCA is sensitive to missing data and poor correlation among water quality parameters (Sarbu and Pop, 2005). Moreover, fuzzy offers simplicity, flexibility, reliable results, can handle incomplete data sets and nonlinear functions. Therefore, Fuzzy has been extensively used in development of decision support system for applications pertaining to water and CPS. This approach have been widely discussed in several environmental applications ranging from development of decision support system based for urban water management (Macropoulas et al., 2003) to Fuzzy based CPS system (Leu and Zhang, 2009)".

6. "User friendly Interface" -> more details required.
   *Reply: Please refer text in reply 1. This text in revised manuscript elaborate the user friendly interface.*

7. Information about detection capabilities is missing: reliability of detection, rate of false positive detections:
   *Reply: Validation method has been changed and earlier mentioned radar chart in fig 3 has been replaced by Mean Average Percentage Error chart as per reviewer_1. The MAPE testify the validity of results can be referred from fig 3 of section Author Comment 1. MAPE can give enough evidence of deviation of proposed system from real and actual values. Therefore, we could measure reliability and rate of false detection from MAPE.*

---

## Author Comment (AC4) · 4 Nov 2017

The overall system cost and comparison analysis ? The cost analysis has been addressed in revised manuscript in section 2.4. overall cost of sensors and hardware unit was 589 US $. However, this cost does not include the cost of consumables, data collection, power source, scientific supervision, labor, resources used for sample collection and shipping to analytical laboratories

b. Does the proposed system is feasible for rural deployments ? Yes, as system is low cost and capable to sustain for longer period of times due to low power requirements.

[Figure]

c. Is is possible to measure fluoride contamination, as author claim that system is adaptable ? Yes, CPS are adaptable and reconfigurable. Therefore, fluoride sensor could also be included in system, if user desired to.

d. Is this proposed system can be integrated with any other sensor array for water? It depends upon type of sensor. Different manufacturers follows the different procedures to develop individual sensor nodes with varied ranges, resolution, requirements weather industrial or potable. The system is primarily designed to collect and process complex data points from sensor nodes and once the data from sensor node is available. System can be integrated as usual.
* * *